# Hair Sample Analysis as a Method of Monitoring Exposure to Bisphenol A in Dogs

**DOI:** 10.3390/ijerph19084600

**Published:** 2022-04-11

**Authors:** Krystyna Makowska, Julia Martín, Andrzej Rychlik, Irene Aparicio, Juan Luis Santos, Esteban Alonso, Sławomir Gonkowski

**Affiliations:** 1Department of Clinical Diagnostics, Faculty of Veterinary Medicine, University of Warmia and Mazury in Olsztyn, Oczapowskiego 14, 10-957 Olsztyn, Poland; andrzej.rychlik@uwm.edu.pl; 2Departamento de Química Analítica, Escuela Politécnica Superior, Universidad de Sevilla, C/Virgen de África, 7, E-41011 Sevilla, Spain; jbueno@us.es (J.M.); iaparicio@us.es (I.A.); jlsantos@us.es (J.L.S.); ealonso@us.es (E.A.); 3Department of Clinical Physiology, Faculty of Veterinary Medicine, University of Warmia and Mazury in Olsztyn, Oczapowskiego 13, 10-957 Olsztyn, Poland; slawomir.gonkowski@uwm.edu.pl

**Keywords:** endocrine disruptors, bisphenol A, toxicology, LC-MS/MS method, canine hair

## Abstract

Bisphenol A (BPA) is an organic substance widely used in the plastics industry. It penetrates food and environment and, as an endocrine disruptor, has detrimental effects on human organisms. Pet animals, which live in the immediate vicinity of humans, are also exposed to BPA; however, knowledge regarding the exposure of dogs to this substance is extremely scarce. This is the first study in which hair analysis has been used to biomonitor BPA in 30 dogs using liquid chromatography and tandem mass spectrometry techniques. The presence of BPA in concentration levels above the method detection limit (1.25 ng/g) was noted in 93.33% of samples. BPA concentration levels were found to range from 7.05 ng/g to 436 ng/g (mean 81.30 ng/g). Statistically significant differences in BPA concentration levels were found between animals with physiological weight and animals with abnormal weight (skinny and obese). In turn, differences between males and females, as well as between young, middle-aged and old dogs, were not statistically significant. The obtained results have clearly shown that hair analysis is a useful method to evaluate the exposure of dogs to BPA. This study also confirmed that dogs are exposed to BPA to a large extent, and this substance may play a role as a pathological factor in this animal species. However, many aspects connected to the influence of BPA on canine health status are unclear and need further study.

## 1. Introduction

Bisphenol A (BPA) (chemical name: 4,4′-(Propane-2,2-diyl)diphenol) is an organic synthetic substance produced during the condensation of acetone and phenol [1,2]. Since products containing BPA are resistant to destruction and light, as well as relatively inexpensive, this substance is commonly used in the plastics industry [2]. BPA is contained in various objects of everyday use, including, among others, bottles, food containers, electronic equipment, household goods, furniture, and even dental fillings [2,3].

It is known that BPA can penetrate plastic, food, soil, and air, as well as drinking, tap, and surface water [4,5,6,7]. Due to the common use of BPA in industry, the presence of this substance has been found in the environment all over the world; it has even been detected in Antarctic ice [8].

BPA also penetrates living organisms through the digestive tract, respiratory system, and skin [2,9,10,11]. Moreover, it is known that prenatal exposure to BPA is also possible because this substance can cross the placenta [12]. Previous publications have described the presence of BPA in human blood, urine, and milk, as well as in domestic and wild animals [13,14,15,16].

Due to its homology to oestrogen, BPA in a living organism binds to estrogenic receptors, leading to disorders in the functions of many internal organs and systems. It is therefore considered a strong endocrine disruptor [2,4]. Previous studies have shown that BPA, among others, has a negative impact on the reproductive, endocrine, nervous, immune, and digestive systems, and causes disturbances to heart and kidney functionality [4,17,18]. BPA also indicates strong hepatotoxic activity. In the liver, BPA causes inflammatory processes, apoptosis and oxidative stress reactions [19], and changes fatty acid and glucose metabolism and intrahepatic innervation [20,21]. A correlation between exposure to BPA and the development and progression of fatty liver disease has been also reported [22,23]. Moreover, some observations have shown a connection between the degree of exposure to BPA and increased risk of diabetes, obesity, and hypertension, as well as neurodegenerative and neoplastic diseases [4,24,25].

It should be pointed out that, contrary to human medicine, the role of BPA as a toxic and disease agent in veterinary medicine has been marginalised until recently. Despite this, it is known that domestic animals, especially dogs and cats, which live together with humans in the same conditions, are exposed to similar environmental pollutants as humans [15,26,27]. Knowledge of both pet animal exposure to BPA and the participation of this substance in pathological processes in such animals is relatively scarce. To date, only a few studies concerning this issue have been published. These studies have reported the presence of BPA in the serum and urine of cats [15,28] and dogs [28,29]. BPA has also been described in canned pet foods [29,30], which confirms that food may be an important factor regarding pet animal exposure to this substance. Moreover, several differences in BPA concentration levels in feline serum have been noted depending on the animal’s age, type of food, and lifestyle (different between indoor cats and cats with outdoor access) [15]. Previous studies have also described that, in pet animals, BPA influences the nervous system and senses [31,32], reproductive system [33], kidneys [34], and selected haematological parameters [15].

To date, matrices other than serum or urine have not been used for monitoring the exposure of dogs to BPA [28,29]. However, in light of recent studies, hair analysis seems to be especially interesting in BPA biomonitoring [35,36]. Hair sample collection is easy and completely non-invasive, in contrast to urine or blood, whose collection may be stressful for animals, especially in restless, aggressive, or skittish individuals. Moreover, hair samples are easy to store and inexpensive to transport, even over long distances. Simultaneously, previous studies have shown that hair analysis is fully usable for monitoring exposure to various environmental endocrine disruptors for both humans [35,37,38,39] and dogs [26,27]. Moreover, it is known that the results obtained from hair analysis are similar—with regard to sensitivity and reliability—to the results obtained by studying blood or urine [40]. Due to the above-mentioned facts, hair analysis seems to be a good alternative to studies involving the classic toxicological matrices used to date (blood, urine), and may become increasingly important in toxicological studies.

Therefore, the aim of the present study was to establish the degree of exposure of dogs to BPA by canine hair analysis. It is the first study in which the analysis of hair samples was used to achieve this goal.

## 2. Materials and Methods

### 2.1. Chemical Reagents

All reagents were of analytical grade, unless otherwise specified. Acetic acid (HAc), ammonium acetate, and sodium dodecylsulphate (SDS) were obtained from Panreac (Barcelona, Spain). HPLC-grade acetone, acetonitrile, methanol, and water were supplied by Romil (Barcelona, Spain). High-purity standard BPA (≥99%) and internal standard BPA-d_14_ (99.5%) were obtained from Sigma Aldrich (Steinheim, Germany) and Dr. Ehrenstorfer (Augsburg, Germany), respectively. A stock standard solution was prepared at 1000 mg/L in MeOH and stored at −18 °C. The working solution was prepared by diluting the stock standard solution in methanol.

### 2.2. Hair Sample Collection

Hair samples were collected from 30 clinically healthy male and female dogs of various breed, age and body condition score (BCS). They were selected randomly from animals whose owners responded to the study announcement, consented to the hair sampling, and provided some information about their animals. According to the information obtained from the owners, all dogs included in the study were indoor dogs living in Olsztyn—a city of about 170,000 inhabitants located in north-eastern Poland—who went out to walk with their owners several (from four to seven) times a day. All dogs included in the study had a similar diet that included mainly canned and dry commercial canine food, and food and water were served in plastic bowls. The number of dogs included in the study was limited by the number of dog owners interested in the study willing to provide a small amount of information regarding their dog.

Since hair sampling is completely non-invasive and the collection of samples was conducted during care treatments, the present study did not require the consent of the Ethical Committee. This is consistent with the Act for the Protection of Animals for Scientific or Educational Purposes of 15 January 2015 (Official Gazette 2015, No. 266), applicable in the Republic of Poland. The exact data concerning dogs included in the experiment are shown in the Appendix A.

Dogs were grouped according to three guidelines: depending on gender, into males (*n* = 9) and females (*n* = 21); depending on age, into young (aged up to 3 years (*n* = 9)), middle-aged (3 to 10 years (*n* = 13)) and old (aged over 10 years (*n* = 8)); and finally, depending on body condition score (BCS), into too thin (BCS 1–3 (*n* = 5)), dogs with correct weight (BCS 4–5 (*n* = 12)), and overweight dogs (BCS 6–9 (*n* = 13). The latter classification was made on the basis of the international canine body condition score system [41].

Hair samples (about 2 g each) were collected from the same place from all dogs. Namely, hair was cut from the abdomen as close to the skin as possible. Immediately upon collection, hair samples were wrapped in aluminium foil and stored in the dark at room temperature until further study. During storage, the hair samples did not contact materials that could contaminate them with BPA.

Before further examination, hair samples were washed to remove surface contaminations. The hair was first washed in ultrapure water, followed by SDS (0.1%, *w*/*v*), and then twice with ultrapure water. In each washing step, the sonication of hair samples for 5 min was made. After washing, the hair was cut into pieces of 2–3 mm long, dried, wrapped in aluminium foil, and stored in the dark at room temperature until further analysis. Three samples of the hair from each dog were subjected to analysis.

### 2.3. BPA Extraction and Analysis of Its Levels

Analysis was performed following the method previously described by Martín et al. [35,38] with slight modifications. Washed hair samples (100 mg) containing BPA-d_14_ (12.5 ng) were incubated with a mixture of methanol and HAc (2 mL, 85:15, *v/v*) in 10 mL glass centrifuge tubes. Incubation lasted for 12 h at 38 °C. The samples were then cooled to room temperature, extracted with 3 mL of acetone in an ultrasonic bath for 15 min, and centrifuged for 10 min at 2900× *g*. The liquid phase was separated into a clean tube and evaporated to dryness under a nitrogen stream at room temperature. The residue was reconstituted in 0.25 mL of methanol, filtered through a 0.22 μm nylon filter. An aliquot (10 μL) of the extract was injected into the instrument.

Liquid chromatography–tandem mass spectrometry analysis was made with an Agilent 1260 Infinity II (Agilent, Santa Clara, CA, USA) and a HALO C-18 Rapid Resolution (50 × 4.6 mm i.d., 2.7 μm particle size) column. The mobile phase was composed of a 10 mM ammonium acetate solution (solvent A) and MeOH (solvent B). The elution gradient was as follows: for 0–14 min, the linear gradient of solvent B rose from 28 to 70% at a flow rate of 0.6 mL/min; this rose from 70% to 80% after 5 min, increased to 100% at 6 min, and then held for 2 min. The column temperature was maintained at 30 °C.

The mass spectrometer was operated in negative electrospray ionisation mode. [M–H]^−^ was used as a precursor ion (m/z 227). The analyses were performed in multiple reaction monitoring (MRM) mode. Two transitions were selected (m/z 227 → 133 for quantification and m/z 227 → 211.8 for confirmation purposes). The other MS parameters were as follows: fragmentor—126 V; collision energy—24 eV; capillary voltage—3000 V; drying gas flow rate—9 L/min; drying gas temperature—350 °C; nebulizer pressure—40 psi. Appendix A shows the BPA mass spectrum.

### 2.4. Background and Quality Control

An important aspect related to BPA analysis is blank contamination owing to the widespread use of plastics and epoxy resins in laboratory materials. As far as possible, plastic materials were avoided in preparing the samples. Moreover, to remove any possible BPA contamination, all glassware were washed with milli-Q water and sonicated with acetonitrile. The material was then rinsed with Milli-Q water and dried in an oven at 120 °C for 4 h.

A quality assurance/quality control protocol was established to guarantee credible and precise results. The protocol contained the use of spiked samples (50 ng/g), BPA standard (20 ng/mL), procedural blanks, and methanol injections into each batch of samples (15). Procedural blanks were processed in the same manner as the hair samples. An average concentration below the method quantification limit (MQL) of BPA was achieved in all analysed procedural blanks. The BPA concentrations in hair samples were calculated by matrix-matched calibration curves. The validation parameters are summarized in the Appendix A. The method detection limit (MDL) and MQL were 1.25 ng/g and 4.2 ng/g, respectively. In the absence of certified reference materials, recovery assays with spiked samples using matrix-matched calibration were used to validate the method. Recovery rates ranged from 94% to 99%, while inter- and intra-day variability was under 6%. Appendix A shows the MRM chromatograms obtained for spiked samples with BPA at low, medium, and high concentration levels and Appendix A shows the MRM chromatogram of a real sample analyzed.

### 2.5. Statistical Analysis

The statistical analysis was performed using GraphPad Prism version 9.2.0 (GraphPad Software, San Diego, CA USA). In the case of a comparison between two groups (male dogs versus female dogs), a non-parametric Mann-Whitney test was used, and in the case of a comparison between three groups (thinner dogs versus dogs with normal weight versus dogs with obesity, and young dogs versus middle-aged dogs versus old dogs) a Kruskal-Wallis test was used. The differences were considered statistically significant at *p* < 0.05.

## 3. Results

In this study, BPA concentration levels were evaluated in 28 out of the 30 dogs included in the experiment (93.33% of analysed samples); in two dogs, the BPA levels were below the method detection limit (MDL). The concentration levels of BPA were different in particular animals and fluctuated from 7.05 ng/g to 436 ng/g with an arithmetic mean of 81.30 ng/g. The data obtained during the present study are summarised in Table 1.

During the present study, the mean concentration levels (±SD) of BPA in male dogs amounted to 58.16 ± 37.01 ng/g, and were slightly lower than those observed in females, which achieved values of 90.56 ± 120.1 ng/g. In the male group, the highest BPA concentration level amounted to 126 ± 1.65 ng/g and was noted in a Poodle, and in the female group it was 436 ± 7.16 ng/g (observed in a Golden Retriever) (Table 1). Nevertheless, there were no intragender statistically significant differences in BPA concentration levels (Figure 1).

Some differences in BPA concentration levels were also noted depending on the age of the animals. In young animals, the mean BPA concentration levels amounted to 60.47 ± 41.78 ng/g and increased with age to 82.98 ± 115.3 ng/g and 108.90 ± 146.7 ng/g in middle-aged and old animals, respectively. In young dogs, the highest concentration level of BPA amounted to 126 ± 1.65 (noted in a Poodle), in middle-aged dogs this amounted 436 ± 7.16 ng/g (noted in a Golden Retriever), and in old dogs this amounted to 307 ± 9.56 ng/g (noted in a Poodle) (Table 1). However, these differences were not statistically significant (Figure 2).

Differences in BPA concentration levels were also noted depending on the weight of the animals. The lowest mean BPA concentration levels, amounting to 26.62 ±20.35 ng/g, were observed in dogs with physiological weight. This value was statistically significantly lower than the mean BPA concentration level noted in thinner dogs (91.08 ± 28.55 ng/g) and dogs with obesity (124.6 ± 137 ng/g) (Figure 3). The highest BPA concentration levels in skinnier dogs amounted to 126 ± 1.65 ng/g (noted in a Poodle), in dogs with physiological weight, 40.1 ± 2.19 ng/g (noted in a mongrel), and in dogs with obesity, 436 ± 7.16 (noted in a Golden Retriever) (Table 1).

## 4. Discussion

The results obtained in this study have shown that dogs are widely exposed to this substance. This is in agreement with previous observations, which have described the presence of BPA in the serum and urine of pet animals (Table 2).

Dogs and cats live in immediate proximity to their owners in the same environment, as a result of which humans and pet animals are exposed to the same environmental pollutants. In animals, just like in humans, one of the main sources of BPA penetrating the body are food and drinking water. It is commonly known that BPA is present in various food products around the world, including fish, meat, milk, and food of plant origin, and the BPA concentration levels clearly depend on the place where the studies have been performed [3,4,42,43]. Moreover, BPA has also been noted in canned food for cats and dogs. In cat food, BPA concentration levels fluctuate from 13 to 136 ng/g [30], and in dog food, depending on the study, from 11 to 206 ng/g [29,30]. In turn, the main cause of BPA presence in canned pet animal food is not the contamination of raw materials used for food production, but the penetration of BPA from the inner layers of cans [29].

The second important source of exposure to BPA may be connected to polluted indoor air and house dust. Previous studies conducted in various world regions have reported that concentration levels of BPA in house dust vary greatly among countries and reach from 9.6 up to 32,000 ng/g [44,45,46]. BPA is also present in indoor air, and its concentration levels depend on the part of the world and degree of industrialisation [47]. The high impact of BPA in indoor dust and air on the degree of pet animal exposure to this substance is confirmed by the fact that the blood serum of indoor cats has been found to contain higher BPA concentration levels than cats with outdoor access [15].

Animal toys, training devices, and bowls may also influence pet animal exposure to BPA. These items are often made from plastic materials containing BPA because in the production of items intended for animals (contrary to items for people) BPA-free plastics are usually not used. However, it is known that BPA may penetrate from canine toys and training devices to the saliva during chewing [48] and be absorbed in the intestine and through the mucosal layer of the mouth [49].

Humans are exposed to BPA in numerous situations which are irrelevant to animals, for example, through contact with thermal paper, electronic equipment, beverages in plastic bottles, and dental fillings [2,50,51]. For these reasons, animals may be less exposed to BPA than humans. However, because exposure to BPA is extremely varied in different parts of the world, this cannot be stated unequivocally. Moreover, many issues are not clear, for example, why BPA concentration levels were much higher in cats than in dogs when samples were taken in the same area [28]. This probably results from (so far unknown) intraspecies differences in BPA metabolism. However, comparing the results from one country, we can see that BPA concentration levels in dogs are lower than in humans. For example, in Poland, concentration levels of BPA in human hair were found to range from 26.1 to 1498.6 ng/g [52], while in dogs they ranged from < MDL to 436 ng/g (this work). However, it should be remembered that studies have been performed in various environments (various cities), which may also affect the obtained results.

A comparison of the results obtained in the present study with previous observations conducted on dogs or cats in other countries (Table 2) is also very difficult since, until now, canine or feline hair samples have not been analysed for BPA content. Previous studies have shown that mean BPA concentration levels in urine and serum did not exceed 23 ng/mL and 3 ng/mL in cats and dogs, respectively (Table 2), and were much lower than those obtained in the present study. This fact may result not only from differences in exposure to BPA and various environmental factors influencing this exposure, but also from extremely varied matrices. It is known that BPA in urine is extracted daily, and BPA in serum is transported to various tissues and urine, while it is unknown exactly how it may accumulate in the hair. Moreover, the analysis of hair takes into account two sources of BPA: (a) external, in which BPA penetrates directly from the environment into the hair; and (b) internal, in which BPA gets inside the hair through the blood and hair follicle/root. These facts may cause BPA concentration levels noted in canine hair to be higher than those noted in urine or blood serum.

During the present study, statistically significantly higher BPA concentration levels were noted in obese and thinner dogs than dogs with physiological weight. Previous studies have reported that BPA is an important obesogenic factor in humans [53,54]. This is connected with BPA influence on adipose tissue and changes in the production and secretion of adipokines, which take part in the regulation of appetite and satiety centres, and are therefore important factors in the pathogenesis of obesity [55,56]. The results obtained in the present study suggest similar BPA activity in dogs, contrary to cats, for whom statistically significant differences in BPA concentration levels between obese animals and animals with physiological weight have not been described [15]. In turn, the reason for the higher BPA concentration levels noted in the present study in thinner dogs is not clear, and may be connected with endocrinal activity disturbances related to insufficient weight.

During the present study, it was noted that BPA concentration levels slightly increased with the age of the animals, but differences between age groups were not statistically significant. In terms of the association between BPA concentration levels and age in humans, the issue is not completely clear. Some studies have shown higher BPA concentration levels in children than adults [39,57], which may be explained by the fact that children are more exposed to BPA in relation to their weight than adults. Moreover, it suggests that metabolic pathways that are not fully developed might play a part in BPA transformation. However, other studies have shown higher BPA concentration levels in adults in comparison with children [35]. Higher concentration levels of BPA in older individuals have also been noted in cats [15], which may be explained by the fact that gastrointestinal disturbances (and therefore changes in absorption and the metabolism of BPA) occur more often in older individuals.

An explanation of the correlation between BPA concentration levels and age is difficult. Although the exact mechanisms of BPA metabolism in particular mammal species are not clear, various factors, including diet, lifestyle, industrialisation, and—in the case of humans—dental fillings or workplace location, may modulate exposure to BPA and therefore the concentration levels of this substance in an organism [2,3,4].

A similar situation concerns intragender differences in terms of concentration levels of BPA. During the present study, statistically significant differences in BPA concentration levels between male and female dogs were not observed, although the value was slightly higher in females. Previous studies conducted on humans have described various observations. Some of them have shown higher BPA concentration levels in men, which, according to the authors, is related to higher androgen levels [58], while others have reported higher concentration levels of this substance in girls than in boys [39]. Other studies, including studies performed on men and women living in the same household, collected on the same day, have not shown intragender differences in BPA concentration levels [59]. Moreover, some studies have reported that, although the levels of total BPA in men and women are similar, intragender differences in levels of BPA metabolites have been found [60]. Intragender differences in BPA concentration levels may be connected to various metabolism rates and hormonal activity in males and females, but also (as in the case of age-dependent differences in BPA levels) with other various factors (for example, diet or dwelling place).

An interesting issue is also the correlation between the degree of exposure to BPA and the breed of the dog. Due to the fact that, in the present study, dogs of various breeds (from one to several individuals of the same breed) were examined, the present results do not clearly answer the question of whether such correlations exist. Clear differences in BPA concentration levels between particular animals within the same breed (for example, in the case of a Schnauzer, where values fluctuated from 7.75 ± 0.65 ng/g to 178 ± 7.40 ng/g) may suggest that exposure to BPA is not correlated with the breed, but with other environmental factors. However, further studies on this issue are necessary, and should involve larger groups of dogs of the same breed.

It should be noted that the analysis of hair for the biomonitoring of BPA concentration levels is a relatively new method, which until now has only been used in a few studies (Table 3).

Both previous studies and the present work have clearly shown that hair sample analysis may be a good alternative to studies using “classic” matrices, such as urine or blood serum, due to its non-invasive character and easiness of sample collection. However, despite its advantages, hair analysis has some limitations. As mentioned above, since studies on hair and hair samples take into account external and internal exposure to BPA, in this method the separation of externally deposited from internally deposited BPA is, therefore, practically impossible. Moreover, hair analysis is not the correct method for studies on short-term fluctuations in BPA concentration levels under rapidly changing factors, because this substance accumulates in the hair over a long time. Therefore, hair analysis is adequate for the monitoring of long-term exposure to BPA, and urine analysis seems to be the best method to study short-term changes.

In spite of these limitations, the results obtained in the present study show that hair analysis may be used in explaining the many aspects associated with the exposure of dogs to BPA. The easy and stress-free collection of hair samples will enable comprehensive studies on large experimental groups concerning the correlation between the degree of exposure to BPA and risk of various pathological states in dogs, e.g., liver and gastrointestinal diseases, diabetes, obesity, endocrinal disturbances, and neoplastic processes. Although information about BPA-induced pathological processes in dogs is rather scarce, previous studies on experimental animals and epidemiological reports on humans [4,17,18,19,20,21,22,23] strongly suggest that BPA may influence both the above-mentioned diseases and other disturbances in dogs. The long-term biomonitoring of dogs’ exposure to BPA using hair sample analysis may also be used in dog breeding, as it may eventually elucidate the influence of BPA on disturbances to the reproductive processes in various canine races. A correlation between the degree of exposure to BPA and the functionality of the canine reproductive system is very likely, because BPA-induced disturbances in reproductive processes in other species are relatively well-known [2,4]. Another problem which may be explained with hair samples analysis is the exposure of dogs to BPA through food, toys, and other equipment involved in dog keeping (such as bowls, collars, leashes). Previous studies strongly suggest that food and/or toys may be a source of BPA that penetrates canine organisms [29,30,48,49]. Moreover, it should be pointed out that this problem concerns not only pet dogs, but also animals working in security services in public and military areas. Therefore, the biomonitoring of BPA in dogs is an important problem in modern veterinary and dog breeding, and the analysis of hair samples (due to easy and stress-free collection) may be one of the methods which will help to solve this problem. Of course, due to very limited previous reports, this issue requires further comprehensive research.

## 5. Conclusions

To conclude, the present study, which may be the first step to further investigations on BPA in pet animal hair, has clearly shown that hair analysis may be useful in determining the degree of exposure of dogs to BPA. The obtained results have also confirmed that dogs are highly exposed to BPA and, hence, this substance may be important as a pathological factor in this species. However, the exact explanation of all aspects connected to BPA influence on the health status of dogs needs further comprehensive and multidirectional study. This might involve evaluating BPA metabolism in dogs, BPA-induced disturbances in internal organ functionality and canine diseases, or the environmental factors affecting the degree of exposure to BPA in dogs.

## Figures and Tables

**Figure 1 ijerph-19-04600-f001:**
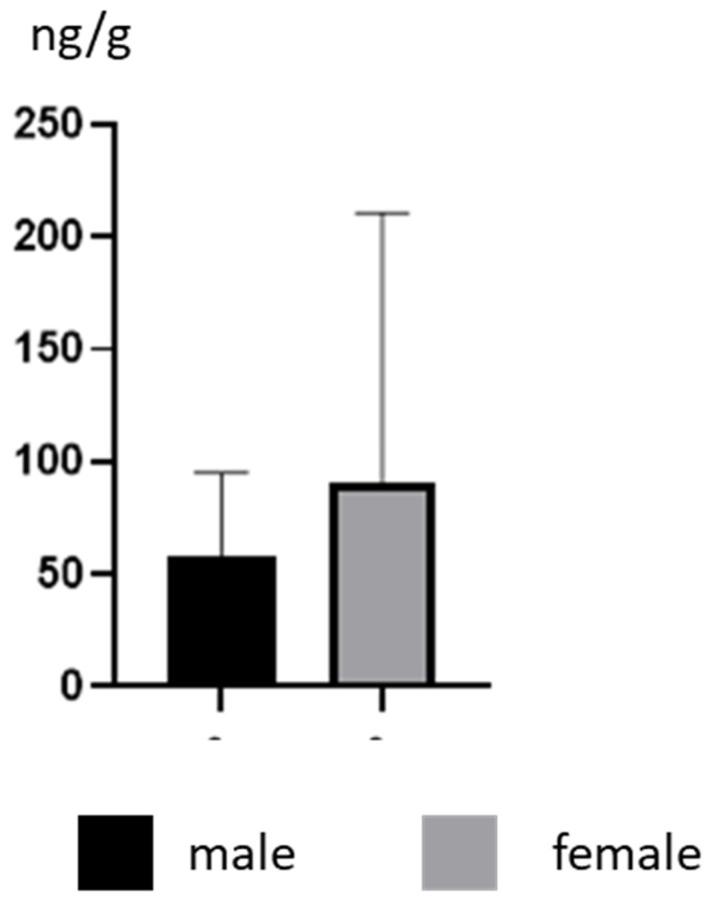
Mean concentration levels (± SD) of BPA in the hair of male and female dogs.

**Figure 2 ijerph-19-04600-f002:**
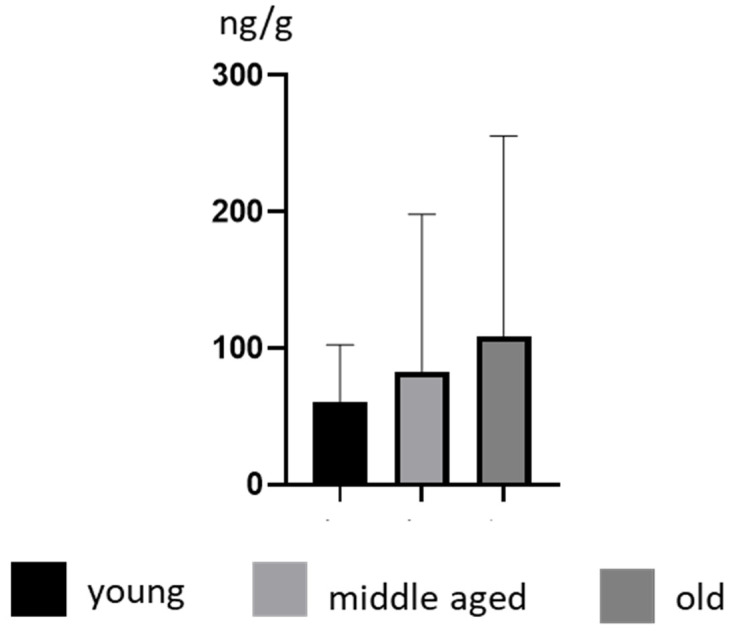
Mean concentration levels (± SD) of BPA in young, middle-aged and old dogs.

**Figure 3 ijerph-19-04600-f003:**
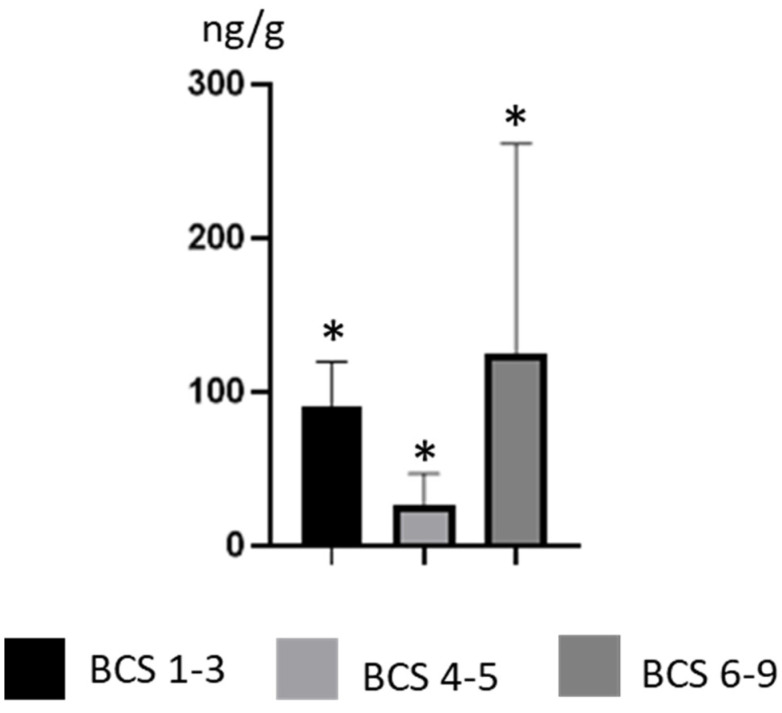
Mean concentration levels (± SD) of BPA in thinner dogs (BCS 1–3), dogs with normal weight (BCS 4–5) and dogs with obesity (BCS 6–9). Statistical differences (*p* ≤ 0.05) are marked with *.

**Table 1 ijerph-19-04600-t001:** Concentration values (ng/g) and frequency of detection of BPA in the canine hair samples (*n* = 30).

Hair Sample Number	Concentration Levels of BPA (ng/g)	
	Geometric Mean (*n* = 3)	Standard Deviation (*n* = 3)
1	307	9.56
2	114	7.49
3	27.2	0.90
4	436	7.16
5	52.4	2.30
6	178	7.40
7	58.6	3.91
8	50.2	1.42
9	<MDL	-
10	13.5	1.23
11	289	60.4
12	80.7	5.76
13	126	1.65
14	14.4	1.28
15	24.3	1.30
16	99.0	14.0
17	31.1	2.98
18	47.6	1.57
19	25.5	1.09
20	10.6	1.62
21	7.85	0.60
21	47.1	1.57
23	<MDL	-
24	7.75	0.65
25	7.05	0.51
26	25.5	2.13
27	74.0	1.17
28	40.1	2.19
29	9.68	0.89
30	72.4	1.32
Cumulative data		
Range (ng/g)	<MDL–436	
Arithmetic mean (ng/g)	81.30	
Geometric mean (ng/g)	43.32	
Median (ng/g)	47.35	
Frequency of detection (%)	93.33	

MDL—Method detection limit = 1.25 ng/g.

**Table 2 ijerph-19-04600-t002:** Previous studies concerning the exposure of pet animals to BPA.

Species	Group of Animals	Mean BPA Levels (ng/mL)	Matrix	References
Cats	All cats included in the study	1.06 ± 0.908	serum	[15]
	Indoor cats	1.27 ± 0.992		
	Cats with outdoor access	06.60 ± 0.529		
	Mature cats (7–10 years old)	1.28 ± 0.994		
	Geriatric cats (over 15 years old	0 420 ± 0.240		
	Cats fed with canned food	1.23 ± 0.935		
	Cats not fed with canned food	0.774 ± 0.795		
	Males	1.14 ± 1.15		
	Females	0.992 ± 0.728		
	All cats included in the study	22.3 ± 155	urine	[28]
Dogs	All dogs included in the study	1.3 ± 4.6		
	Dogs before feeding with the canned food containing BPA	0.7 ± 0.15	serum	[29]
	Dogs after 14 days of feeding with canned foods containing BPS	2.2 ± 0.15		

**Table 3 ijerph-19-04600-t003:** Biomonitoring studies of BPA in the hair.

Species	Country	Number of Samples	BPA Concentration Levels (ng/g)	Reference
Human	Belgium	114	<LOQ–587.1	[61]
	Greece	69	13.1–192.8	[37]
	Greece	122	2.6–205	[39]
	Greece	100	9.6–650.3	[36]
	Korea	10	17–22.9	[62]
	Spain	6	24–158	[38]
	Spain	6	9.2–45	[63]
	Spain	42	24.4–1427	[35]
	Poland	42	26.1–1498.6	[52]
Grey seal	Poland	17	<LOQ–137.2	[52]

MDL—method detection limit; LOQ—limit of quantification.

## Data Availability

Data is contained within the article or Appendix A.

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
