# Peer review of "Hair Sample Analysis as a Method of Monitoring Exposure to Bisphenol A in Dogs"

_ijerph, 2022, doi:10.3390/ijerph19084600_

Round 1

Reviewer 1 Report

The paper submitted for review entitled "Hair samples analysis as a method of monitoring of exposure to bisphenol A in dogs" is interesting and addresses the issue of canine exposure to BPA and indicates the need to clarify many aspects related to the effects of BPA on the health status of dogs, which still requires much research.
I provide some comments and suggestions on the manuscript text:
Line 28: Is it not worth including "hair of dogs" in the keywords instead of "fur"?
Line 95: In the introduction of the paper, the authors use the word "hair", followed by the use of the word "fur" in the Materials and methods section - I suggest that this be standardized, especially as there is no need to repeat the word too often in the text of the paper, as well as in Lines: 176, 184, 305, 307 reference is made to hair samples
Line 137: Please expand the description of methodology in the context of LC-MS/MS analysis, too laconically
Line 318: I suggest to edit this part of the paper in the context of informing about the decision related to the use of a small experimental group to achieve the aim of the study and not to indicate as a limitation of the study performed
Line 453-454: Correct the notation of the literature entry.

Author Response

The authors thank the Reviewer for evaluation of the manuscript. All suggestions have been taken into account.:

  • the term “canine hair” has been added in keywords (line 28)
  • the term “hair” has been standardized in whole manuscript. The term “fur” has been replaced by “hair”
  • The description of methodology has been expanded:

“The elution gradient was as follows: 0-14 min, linear gradient from 28 to 70 % of sol-vent B, at a flow rate of 0.6 mL/min, from 70 % to 80 % of solvent B in 5 min, and then increased to 100 % in 6 min and held for 2 min. The column temperature was main-tained at 30 ºC.

The mass spectrometer was operated in negative electrospray ionisation mode. [M–H]− was used as precursor ion (m/z 227). The analyses were performed in multiple reaction monitoring (MRM) mode. Two transitions were selected (m/z 227  133 for quantification and m/z 227  211.8 for confirmation purposes). Other MS parameters were as follows: fragmentor, 126V; collision energy, 24 eV; capillary voltage, 3000 V; drying gas flow rate, 9 L/min; drying gas temperature, 350 °C; nebulizer pressure, 40 psi.”(lines 147-156)

And

“Procedural blanks were processed in the same manner as the hair samples. An average concentration below the method quantification limit (MQL) of BPA was achieved in all analysed procedural blanks.” (lines 167-169)

  • According to the suggestion of the Reviewer the fragment about limitations at the end of the manuscript has been replaced by the fragment about decision about relatively small experimental group in materials and methods: “The number of dogs included in the study was limited by the number of dog owners interested in the study and willing to provide short information regarding the dog.” (lines 108-110)
  • Notation has been corrected (lines 556-557)

The authors hope that improvements will allow to publish the manuscript in the International Journal of Environmental Research and Public Health

Reviewer 2 Report

The paper by Makowska et al. deal with an interesting topic, the use of hair analysis as a simple and  non-invasive method of monitoring of BPA contamination in dogs. The paper is well-written and understandable, and the results are correctly interpreted. I only suggest to the Authors to insert in the keywords the word “dog” or “canine fur” to help readers understand that the paper is about dogs. English writing needs only minor changes, changes that are usually made during the publication process.

Author Response

The authors thank the Reviewer for positive review  of the manuscript.

  • The term “canine hair” has been added into keywords
  • Manuscript has been corrected by a native speaker

Reviewer 3 Report

The manuscript “Hair samples analysis as a method of monitoring of exposure to 2 bisphenol A in dogs” by Makowska et al. is a well-organized and targeted paper by an established group. The specific paper targets the question whether a LC-MS/MS method can be used to determine bisphenol A concentrations in dog hair samples and whether bisphenol A uptake levels correlate with body weight. There are, however, some major points that need the authors’ attention:

  1. A paragraph should be added, describing the added value of biomonitoring bisphenol A in dog hair samples in respect to:
  • Longtime monitoring of the bisphenol A concentration
  • Correlation studies of bisphenol A content and various diseases or adverse health conditions in dogs
  • Correlation studies with reproductive success in different dog species
  • Correlation studies with different dog food sources or with dogs using different animal toys, dog training devices in dogs, which are used for security services in public and military areas.

This would highlight the potential of the methodology and significantly contribute to the visibility and overall acceptance of the paper within the scientific community

2. A table should be included, where the bisphenol A concentration is mentioned for each of the 30 dog individuals. It should be added, whether only one sample was analysed for each dog or whether triplicates were used for the quantification of bisphenol A for each individual dog and the mean of the triplicates was used.

3. Due to the high number of different dog species for each comparison in figure 1, 2 and 3 the highest absolute values should be stated for individuals from the female group in figure 1, the middle aged and old group in figure 2 and the BCS 6-9 group in figure 3. A statement should be made, whether the highest concentrations occurred in dog species, where only one single dog of this species was tested within the experiment or whether there were other individuals tested with significant lower bisphenol A amounts. These results should be idiscussed in the discussion section.

4. Page 4, line 155: For the procedural blanks it should be stated, a) whether there were procedural blanks, which exceeded the LOQ (or MQL) and if there were any, how many it where and what the highest concentration found was in comparison to the MLQ or LOQ.

5. Page 2, line 50: liver toxicity of bisphenol A needs to be included here.

6. Page 3, line 139-140: the HPLC separation was isocratic, that should be added somewhere in the method description.

7. The text needs revision of the English language.

Author Response

The authors thank the Reviewer for insightful review of the manuscript. The Reviewer's comments allowed the manuscript to be significantly improved. All suggestions have been taken into account.:

1) According to suggestions of the Reviewer, fragments about value of biomonitoring bisphenol A in dog hair in respect to long-time biomonitoring of BPA in various dogs breeds, correlation between exposure to BPA and various dogs diseases and reproductive disturbances and sources of BPA have been added although the knowledge about these issues is extremely scanty:

  • lines 378-386: “An interesting issue is also the correlation between the degree of exposure to BPA and the breed of the dog. Due to the fact that in the present study, dogs of various breeds (from one to several individuals of the same breed) were examined, the present results do not clearly answer the question of whether such correlations exist. Clear differences in BPA concentration levels between particular animals within the same breed (for example, in the case of a schnauzer, where values fluctuated from 7.75±0.65 ng/g to 178±7.40 ng/g) may suggest that exposure to BPA is not correlated with the breed, but with other environmental factors. However, further studies on this issue are necessary involving larger groups of dogs of the same breed.”

Lines 403-426: “In spite of these limitations, results obtained in the present study show that hair analysis may be used in explaining the many aspects associated with exposure of dogs to BPA. Easy and stress-free collection of hair samples enables comprehensive studies on large experimental groups concerning the correlation between the degree of expo-sure to BPA and risk of various pathological states in dogs, e.g. liver and gastrointestinal diseases, diabetes, obesity, endocrinal disturbances and neoplastic processes. Although information about BPA-induced pathological processes in dogs is rather scarce, previous studies on experimental animals and epidemiological reports on humans [4,17-23], strongly suggest that BPA may influence both the above-mentioned diseases and disturbances in dogs. Long-time biomonitoring of dogs’ exposure to BPA using hair sample analysis may also be used in dog breeding to eventually elucidate the in-fluence of BPA on disturbances of reproductive processes in various canine races. The correlation between the degree of exposure to BPA and the functionality of the canine reproductive system is very likely because BPA-induced disturbances in reproductive processes in other species are relatively well-known [2,4]. Another problem which may be explained with hair samples analysis is the exposure of dogs to BPA through food, toys and other equipment involved in dog-keeping (such as bowls, collars, leashes). Previous studies strongly suggest that food and/or toys may be a source of BPA penetrating into canine organisms [29,30,48,49]. Moreover, it should be pointed out that this problem concerns not only pet dogs but also animals working in security services in public and military areas. Therefore, biomonitoring of BPA in dogs is an important problem in modern veterinary and dog breeding, and analysis of the hair samples (due to easy and stress-free collection) may be one of the methods which will help to solve this problem. Of course, due to very limited previous reports, this issue requires further comprehensive research.”

2) A table with values from all dogs included into the study has been added (Table 1). Moreover, the phrase “Three samples of the hair from each dog were subjected to analysis. “ (lines 132-133) has been added.

3) Information about highest values in each group and about breeds, in which these values were noted has been added. The Reviewer suggested to add mentioned above information on figures. In opinion of the authors adding this information to graphs would make them less readable. So, information has been added into the text:

  • a) 250-252: “In the male group, the highest BPA concentration level amounted to 126±1.65 ng/g and was noted in a poodle, and in the female group it was 436±7.16 ng/g (observed in a golden retriever) (Table 1).”
  • b) lines 260-262: “In young dogs, the highest concentration level of BPA amounted to 126±1.65 (noted in poodle), in middle-aged dogs 436±7.16 ng/g (noted in golden retriever) and in old 307±9.56 ng/g (noted in poodle) (Table 1).”
  • c) lines 270-273: “The highest BPA concentration levels in skinner dogs amounted to 126±1.65 ng/g (not-ed in a poodle), in dogs with physiological weight 40.1±2.19 ng/g (noted in a mongrel) and in dogs with obesity 436±7.16 (noted in a golden retriever) (Table 1).”

Moreover, following fragments have been added: “An interesting issue is also the correlation between the degree of exposure to BPA and the breed of the dog. Due to the fact that in the present study, dogs of various breeds (from one to several individuals of the same breed) were examined, the present results do not clearly answer the question of whether such correlations exist. Clear differences in BPA concentration levels between particular animals within the same breed (for example, in the case of a schnauzer, where values fluctuated from 7.75±0.65 ng/g to 178±7.40 ng/g) may suggest that exposure to BPA is not correlated with the breed, but with other environmental factors. However, further studies on this issue are necessary involving larger groups of dogs of the same breed.” (lines 378-386).

4) and 6)  Additional information about methodology has been added:

Lines  147-156: “The elution gradient was as follows: 0-14 min, linear gradient from 28 to 70 % of sol-vent B, at a flow rate of 0.6 mL/min, from 70 % to 80 % of solvent B in 5 min, and then increased to 100 % in 6 min and held for 2 min. The column temperature was maintained at 30 ºC. The mass spectrometer was operated in negative electrospray ionisation mode. [M–H]− was used as precursor ion (m/z 227). The analyses were performed in multiple reaction monitoring (MRM) mode. Two transitions were selected (m/z 227  133 for quantification and m/z 227  211.8 for confirmation purposes). Other MS parameters were as follows: fragmentor, 126V; collision energy, 24 eV; capillary voltage, 3000 V; drying gas flow rate, 9 L/min; drying gas temperature, 350 °C; nebulizer pressure, 40 psi.”

Lines 167-169: “Procedural blanks were processed in the same manner as the hair samples. An average concentration below the method quantification limit (MQL) of BPA was achieved in all analysed procedural blanks.”

5) information about liver toxicity of BPA has been added: lines 51-55: “BPA also indicates strong hepatotoxic activity. In the liver, BPA causes inflammatory processes, apoptosis and oxidative stress reactions [19] and changes fatty acid and glucose metabolism and intrahepatic innervation [20,21]. A correlation between exposure to BPA and the development and progression of fatty liver disease has been also reported [22,23].”

6) Manuscript has been corrected by a native speaker in English

The authors hope that improvements will allow to publish the manuscript in the International Journal of Environmental Research and Public Health.

This manuscript is a resubmission of an earlier submission. The following is a list of the peer review reports and author responses from that submission.